# Mechanisms of NURR1 Regulation: Consequences for Its Biological Activity and Involvement in Pathology

**DOI:** 10.3390/ijms241512280

**Published:** 2023-07-31

**Authors:** Ángel Juan García-Yagüe, Antonio Cuadrado

**Affiliations:** 1Department of Biochemistry, Medical College, Autonomous University of Madrid (UAM), 28029 Madrid, Spain; antonio.cuadrado@uam.es; 2Instituto de Investigaciones Biomédicas “Alberto Sols” (CSIC-UAM), 28029 Madrid, Spain; 3Instituto de Investigación Sanitaria La Paz (IdiPaz), 28027 Madrid, Spain; 4Centro de Investigación Biomédica en Red de Enfermedades Neurodegenerativas (CIBER-CIBERNED), Av. Monforte de Lemos, 3-5. Pabellón 11, Planta, 28029 Madrid, Spain

**Keywords:** NURR1, nuclear-cytoplasmic distribution, heterodimerization, phosphorylation, SUMOylation

## Abstract

NURR1 (Nuclear receptor-related 1 protein or NR4A2) is a nuclear protein receptor transcription factor with an essential role in the development, regulation, and maintenance of dopaminergic neurons and mediates the response to stressful stimuli during the perinatal period in mammalian brain development. The dysregulation of NURR1 activity may play a role in various diseases, including the onset and progression of neurodegenerative diseases, and several other pathologies. NURR1 is regulated by multiple mechanisms, among which phosphorylation by kinases or SUMOylation are the best characterized. Both post-translational modifications can regulate the activity of NURR1, affecting its stability and transcriptional activity. Other non-post-translational regulatory mechanisms include changes in its subcellular distribution or interaction with other protein partners by heterodimerization, also affecting its transcription activity. Here, we summarize the currently known regulatory mechanisms of NURR1 and provide a brief overview of its participation in pathological alterations.

## 1. Introduction

Nuclear receptor-related factor 1 (NURR1, also known as NR4A2) is an orphan nuclear receptor initially characterized as a transcription factor with critical functions in the development, differentiation, maintenance, and survival of dopaminergic (DAergic) neurons in the midbrain [1], as well as the olfactory bulb, hippocampus, and cortex [2,3]. It is a member of the family of ligand-activated transcription factors known as nuclear receptors. It belongs to the NGFI-B subfamily of transcription factors, which also includes NUR77 (NR4A1, NGFI-B), and NOR-1(NR4A3), the three of them having a high degree of conservation in their DNA-binding domain (DBD) with ≈91–95% similarity, or ligand-binding domain (LBD) with ≈60% similarity [3]. The N-terminus of NURR1 contains an activation function-1 (AF-1) domain, which mediates ligand-independent transactivation [4]. The DBD is responsible for recognizing the response element and binding to the promoter. The C-terminus also comprises the LBD and the ligand-dependent activation function-2 (AF-2) domains (Figure 1) [4,5]. A characteristic of this transcription factor is that, unlike most other nuclear receptors, it lacks a hydrophobic pocket for ligand binding. Thus, it can function as a ligand-independent nuclear receptor [6,7]. NURR1 can bind to specific DNA binding sites. As a monomer binds to the nerve growth factor-induced clone B (NGFI-B) response element (NBRE; 5′-AAAGGTCA-3′) [8]. As a homodimer or heterodimer binds to the Nur response element (NurRE; 5′-TGACCTTT-n6-AAAGGTCA-3′) [4,9], and has the capacity to function as a constitutively active transcription factor. NURR1 also heterodimerizes with retinoid X receptor (RXR) and binds to the direct repeat element with the five spacer nucleotide element (DR5; 5′-GGTTCA-n5-AGGTCA-3′) to regulate cell proliferation and survival [10]. NURR1 regulates the expression of several dopamine (DA) metabolic genes in the midbrain [3,11,12] and other non-DAergic genes as neurophilin-1 [13] or RET (GDNF receptor) [14]. Although the mechanisms governing NURR1 activity are poorly known, some of them are mainly related to the control of its stability [15,16,17], the change in its subcellular localization [18], or heterodimerization with specific proteins [19,20,21]. The post-transductional modification includes the addition of Small Ubiquitin-like Modifier (SUMO) proteins or SUMOylation (Figure 1) [22,23], as well as phosphorylation (Figure 1) by kinases that mark NURR1 for subsequent degradation by the ubiquitin-proteasome system (UPS) [24]. In this review, we will summarize the evidence for the role of NURR1 in several diseases, and the current knowledge about the mechanisms that regulate its transcriptional activity.

## 2. NURR1 in Pathology

The most compelling effect of NURR1 dysregulation is its association with the onset and progression of Parkinson’s disease (PD) [25,26,27]. Genetic ablation of Nurr1coding gene in mice revealed that this gene is needed for the development of DA neurons in the midbrain [28]. NURR1-deficient cells initially express certain DA neural markers, though they cannot innervate their forebrain targets. NURR1 is still present in postnatal DA neurons, and accumulating evidence suggests that it is essential for the maintenance of the DAergic phenotype of these cells. Moreover, NURR1 is also related with other neuropathologies, such as Alzheimer’s disease (AD) [29,30,31]. NURR1 may act as an essential regulator of hippocampal functions, such as synaptic plasticity and cognitive functions [32,33,34], in addition to being an essential mediator of neuroprotection and anti-inflammatory responses to neuropathological stress [35,36]. Interestingly, several studies indicate that there are altered levels of NURR1 in nerve cells treated with Aβ1-42, in AD animal models, and in the brains of AD patients [25,37,38].

A recent study has shown that NURR1-expression is regulated in ischemic stroke [39]. These effects were detected in acute ischemia by middle cerebral artery occlusion (MCAO)/reperfusion in a rat model. NURR1 overexpression inhibited tumour necrosis factor-α (TNF-α) levels in microglia, alleviated infarct volume, and improved the neurological outcomes in an acute stroke model [40].

NURR1 is also implicated in schizophrenia [41,42]. Thus, *Nurr1*^+/−^ heterozygous mice exhibited behavioural patterns associated with symptoms of schizophrenia, suggesting a potential animal model [32,41]. Furthermore, protein and mRNA expression levels of NURR1 were reduced in the prefrontal cortex of schizophrenia patients [43]. Whereas the abnormal functioning of DAergic neurons in the brain cortex and subcortical areas is associated with schizophrenia [42], the correlation with NURR1 expression needs to be further investigated in this pathology.

The role of NURR1 in drug addiction is controversial because it seems to be related to the duration of exposure and the drug type. A few studies have reported that addictive drugs, such as cocaine and heroin, reduce NURR1 transcript levels in the midbrain [44,45].

Attention deficit hyperactivity disorder (ADHD) impairs attention and leads to locomotor hyperactivity. The affected brain location, the Ventral Tegmental Area, exhibits a robust NURR1 expression [46,47]. Furthermore, decreased synaptic DA markers have been reported in the DA reward pathway in ADHD patients. A study was conducted in heterozygous for *Nurr1*^+/−^ mice, using prenatal immune activation as a model for attention impairment, and it was reported that genetic and environmental factors synergistically affect attention impairment and locomotor hyperactivity [47].

NURR1 is also implicated in some non-neural pathologies such as several cancers [48,49,50] atherosclerosis [51], and inflammation [52]. In cancer, NURR1 can promote or suppress malignant progression depending on the cellular context. Cancer cell proliferation, invasion, and anchorage-independent growth are inducible by NURR1 overexpression and hyper-activation [53,54]. It also enhances cell survival by suppressing apoptosis [55,56]. NURR1 enhances cancer aggressiveness and provides therapeutic resistance to radiotherapy and chemoresistance to 5-fluorouracil [57,58].

## 3. Regulation of NURR1 Activity by Protein-Protein Interactions

Protein-protein interactions control many physiological processes such as transcriptional control, transition among cytoplasm and nucleus, degradation, etc.; the dysregulation of these processes may result in a disease state. NURR1 binds to specific DNA-binding sites in the vicinity of regulated genes and can recognize DNA as a monomer, homodimer, or heterodimer with the heterodimerization partner RXRα, and RXRγ, but not with RXRβ (Table 1) [21,59,60]. The heterodimers recognize a DNA-binding site that is composed of two consensus NR-binding motifs, organized as direct repeats, that are separated by five nucleotides (DR5) [61,62]. RXR is activated by its ligands in the RXR-NURR1 heterodimer [59,63]. Thus, natural RXR ligands, such as 9-cis-retinoic acid or docosahexaenoic acid, may impact NURR1 activity [64]. The I-box domain of NURR1 is crucial for heterodimerization with RXR, and subsequent binding to the response elements DR5 and NBRE (Table 1) [30,65,66]. Recently, specific endogenous [29,67] or synthetic ligands [68] for RXR-NURR1 heterodimers have been identified in the brain, which help to identify new functional target genes. As reported by Volakakis et al. in 2015, the specific RXR ligand bexarotene has provided promising neuroprotective effects in PD models [69]. They showed that RXR ligands could increase neurotrophic signalling over the neurotoxicity action of the α-SYN, but provided a mixed picture of its potential in a PD rat model. Other RXR ligands with neuroprotective effects are HX600, IRX4204, and BRF110 [70,71,72]. As shown by Loppi et al. in 2018, HX600 reduced microglia-expressed pro-inflammatory mediators and prevented inflammation-induced neuronal death in co-culture of neurons and microglia [70]. In ischemic mice, HX600 protected endogenous microglia from ischemia-induced death and prevented leukocyte infiltration. In addition, the IRX4204 and BRF110 RXR ligands promote the survival and maintenance of nigral DAergic neurons in primary mesencephalic cultures from PD mice and in pluripotent stem cells (iPSC)-derived DAergic neurons from PD patients. The oral administration of IRX4204 activates NURR1 and attenuates neurochemical and motor deficits in a rat model of PD.

NURR1 also interacts with the glucocorticoid receptor (GR) through the DBD or the N-terminal NURR1 domain (Table 1) [19,65,81]. Carpentier et al. demonstrated that both nuclear receptors co-localize to the hippocampus and the substantia nigra of the mouse brain [19]. Several GR domains appear to associate with NURR1. The GR-induced increase in NURR1 transcription activity requires the N-terminal domain of GR, but not a functional DNA-binding domain. On the other hand, the NURR1 domain that is involved in GR interaction has not been conclusively identified. Of note, GR transcriptional activity in At20 cells is inhibited by NURR1, while in other cell types such as PC12 cells, GR increases activity of NURR1 in a dexamethasone-dependent manner [19,65].

Another well-known and essential partner of the NURR1 is the homeobox transcription factor PITX3 (Table 1) [73,82]. NURR1 alone cannot drive the DAergic phenotype in meso-diencephalic DAergic neurons, due to co-repression by PSF (SFPQ; PTB-associated splicing factor) and SMRT (silencing mediator of retinoic acid and thyroid hormone receptor), that can interact with NURR1 and occupy its promoters target genes (Figure 2). SMRT is a co-repressor that can bind to unliganded nuclear receptors, and is believed to keep these complexes in a repressed state [61]. In the absence of PITX3, NURR1 is inhibited by interaction with the co-repressor SMRT, and the recruitment of PITX3 decreases the interaction with SMRT. PITX3 acts through histone deacetylases (HDACs) to maintain promoters in a deacetylation-repressed state. In fact, the interference of HDAC-induced repression in *Pitx3*^−/−^ mouse embryos effectively rescue the expression of NURR1 target genes, bypassing the need for PITX3 [73]. Hence, PITX3 is essential for the transcriptional activity of NURR1, activating its function through its response elements and allowing the development of the DAergic phenotype. This mechanism provides new insights underlying the development of midbrain DAergic neurons and stem cell reprogramming as future therapies for PD [82].

p57Kip2 (cyclin-dependent kinase inhibitor 1C (CDKN1C)) is another protein that makes heterodimers with NURR1 and is essential for normal DAergic neuron development (Table 1) [74]. Similar to the p57Kip2/LIM kinase1 interaction, p57Kip2 binds the N-terminal domain of NURR1 to inhibit its activity [74,83]. Furthermore, LIM-domain-containing proteins bind to the Sin3A-histone deacetylase co-repressor complex, and NURR1 interacts directly with Sin3A [73].

The AF1 domain of the NURR1 can bind to the steroid receptor co-activators (SRCs) (Table 1) [75]. Indeed, the SRC1 PAS-B domain can directly interact with the AF1 domain of the NR4A receptor NURR1. In this regard, the NURR1 LBD/AF2 domain can be covalently regulated by some DA and prostaglandin A1 (PGA1) [64] and prostaglandin A2 (PGA2) [84] metabolites that increase NURR1 transcriptional activity through an unknown mechanism, and lead to the recruitment of SRC1 and SRC3 [64].

Another group of proteins that interact with NURR1, and had been previously described to interact with NUR77, is the peptidyl-prolyl isomerase PIN1 and FHL2 (Table 1). PIN1 was identified in a dual-hybrid yeast screen and showed interaction with NURR1 N-terminal and DBD domains. PIN1 improves the transcriptional activity of NURR1 without changing its protein stability, unlike the protein stability of NUR77, which is increased by PIN1 [76,85]. On the same screen, the LIM-only domain protein FHL2 was identified and as observed for NUR77, FHL2 binds the N-terminal domain and DBD of NURR1 and inhibits its activity [77].

NURR1 is involved in the expression of Forkhead transcription factor FOXP3, which is responsible for the differentiation of regulatory T cells (Treg) [86]. The control of gene expression in CD4^+^ cells is ensured by the direct interaction of NURR1 with RUNX1 (Table 1). In macrophages and microglia, an anti-inflammatory function has been attributed to NURR1 involving inhibition of NF-κB p65 [76]. In fact, the interaction between NURR1 and NF-κB p65 requires p65 to be phosphorylated at Ser486. Subsequently, NURR1 is directly linked to the co-repressor complex with CoREST (Table 1) that has been recruited and binds directly to NURR1 [87]. It is the NURR1/CoREST-mediated transrepression complex that blocks NF-κB p65 activity [76].

It had been reported that NURR1 has antiapoptotic functions. NURR1 overexpression decreases BAX expression, whereas NURR1 knockdown increases its expression. These effects are related to physical interaction with p53 (Table 1), repressing its apoptosis and cell-cycle arrest functions [80]. This interaction of NURR1 is performed between its DBD domain and the carboxyl-terminal of p53. Thus, NURR1 represses p53 transcriptional activity in interaction-dependent and dose-dependent manners, promotes cell survival, and implicates an important role in carcinogenesis and other diseases [80].

## 4. Regulation of NURR1 by Phosphorylation

The abundant serine (59), threonine (28), and tyrosine (20) residues in NURR1 human provide potential phosphorylation sites for various protein kinases (Figure 1). These modifications may affect the interaction between NURR1 and other proteins that impact its stability and interaction with the NBRE sequence.

Several reports describe phospho-sites in the NURR1 paralogue NUR77 that can be potentially extrapolated to NURR1 [67,68]. In 2004, Nordzell et al. were the first to describe a specific NURR1 region regulated by phosphorylation located between amino acids 124 and 133 (124-PSSPPTPSTP-133) [88]. This stretch contains several Ser/Pro and Thr/Pro amino acids that conform the classical sequence targeted for phosphorylation by the MAP-kinase family, including ERK1 and ERK2 (PX(S/T)P) [89,90]. Using a luciferase reporter assay in combination with multiple NURR1 deletion- or point-mutants linked to Gal4 in this region, Nordzell et al. identified phosphorylation dependent sites that regulate its transcriptional activity, presumably regulated by MAP-kinases [89]. One issue with this experimental approach is the potential artefactual character of these Gal4 chimeras. Three years later, Zhang et al. performed a “pull-down” assay that purifies GST-tagged recombinant NURR1 interacting with ERK1/2, into four potential ERK docking sites [89,91]. They also identified Ser126 and Thr132 as the ERK1/2 targets in an in vitro assay [83] (Table 2) [91]. Although the effect of ERK1/2 on NURR1 phosphorylation is compelling, the impact of this phosphorylation on NURR1 transcriptional activity seems controversial [91]. Thus, Jacobsen et al. showed in 2008 that a PD-related NURR1 Ser125Cys mutant exhibited lower transcriptional activity towards a luciferase reporter and that the Ser was phosphorylated in an ERK1/2-depedent manner [44]. However, according to the consensus motif for MAP-kinase phosphorylation (124-P***S***SPPTPSTP-133), Ser 126 instead of Ser 125 should be the one phosphorylated.

Lu et al. reported that the protein deglycase DJ-1, also known as Parkinson’s disease protein 7 (PARK7), enhanced the nuclear translocation of NURR1 and its transcriptional activity, in the DAergic cell line MN9D [79]]. According to the authors, these biological activities of NURR1 were regulated through ERK1/2 [94,95], and a PD-related L166P DJ-1 mutant abolished this effect.

Although NURR1 can be regulated by ERK1/2 through the control of its gene expression [94,96], the direct effect of these kinases on NURR1 phosphorylation in vivo is not compelling yet. In fact, it is possible that NURR1 could be phosphorylated by other kinases downstream of the MAP-kinases pathway. Thus, RSK (ribosomal S6 kinase) and MSK (mitogen- and stress-activated kinase) can phosphorylate Ser354 of the other orphan receptor family NUR77 [86,93], and both kinases are regulated by ERK1/2. In parallel, the authors show that other proteins besides NUR77, such as NURR1 and NOR-1, are also phosphorylated by this kinase (Table 2) [97]. In this way, the phosphorylation of NURR1 via ERK1/2 might result through RSK- or MSK-mediated phosphorylation at the equivalent Ser347, because both kinases are directly regulated by ERK1/2. Curiously, the Ser347 regulated by RSK/MSK are localized into the second nuclear localization signal (NLS2) of NURR1 [93], suggesting active participation in the control of its subcellular localization.

ERK5 is a MAP-kinase that phosphorylates NURR1 in Thr168 and Ser177 [83] (Table 2). NURR1 Thr168Ala and Ser177Ala point mutations were not responsive to ERK5-mediated activation of a luciferase reporter containing three tandem sequences of the NURR1-responsive NBRE enhancer. Besides, ERK5 binds the N-terminal domain of NURR1, where both target residues are located, as well as the C-terminal ligand-binding domain (LBD). However, the effect of ERK5 phosphorylation is unclear and might lead to increased stability, change of subcellular localization, heterodimerization with other protein partners, or increased binding to DNA. The connexion between ERK5 and NURR1 might be implicated in the protection and maintenance of the DAergic phenotype when DAergic cells are exposed to toxic stimuli such as 6-OHDA or α-SYN aggregates [98,99].

Although the motif (124-PSSPPTPSTP-133) of NURR1 was initially correlated with possible phosphorylation and regulation by ERK1/2 (MAP-Kinase), recently it was found that GSK-3β can also phosphorylate and regulate NURR1 stability through the same motif (Table 2) [99]. García-Yagüe et al. (2021) showed that activation of GSK-3β by toxic α-SYN leads to phosphorylation and subsequent UPS degradation of NURR1, leading to loss of DAergic markers [16]. The relationship between α-SYN and NURR1 was previously described in Yuan et al. (2008), who reported that overexpression of the toxic form α-SYN regulates the expression levels of NF-κB and increases the activity of glycogen synthase kinase 3β (GSK-3β) in DAergic neurons, which would have adverse effects on the NURR1 levels [16,100,101]. Mechanistically, this group identified the putative “MAP-Kinase’s motif” between (124-PSSPPTPSTP-133) amino acids, harbouring proline-directed serine residues, as the main target of this kinase. GSK-3β is a kinase that phosphorylates its substrates in two specific consensus sequences, (Ser/Thr)-Pro or (Ser/Thr)-X3-(pSer/pThr), where X is any residue [102]. NURR1 contains at least five putative sequences with serines or threonines that conform to the consensus motif for GSK-3β phosphorylation, including the so-called “Core 2” by the authors [16]. Although the authors did not demonstrate that NURR1 is phosphorylated directly by GSK-3β, this kinase can regulate NURR1 stability through the Core 2 motif because the authors show a NURR1 band-shift change in 1D electrophoretic gels SDS-PAGE or 2D gels. Moreover, the point mutant in Core 2 renders NURR1 insensitive to degradation when GSK-3β-induced degradation occurs, therefore, protecting it against α-SYN toxicity in a PFF treatment cell model. In fact, it is conceivable that Core 2 functions as a molecular switch that coordinates NURR1 activation or proteasomal degradation in response to the MAP-kinase of GSK-3 in coordination with others signals [43]. Moreover, many GSK-3β substrates must be previously phosphorylated by another kinase [69,70,91]. Therefore, it is also possible that initial phosphorylation, originated by ERK, might lead to transient NURR1 activation and, at the same time, the ERK-phosphorylated NURR1 would prime this protein for further GSK-3β-mediated phosphorylation and degradation at a later stage when cell signaling is reduced and GSK-3β becomes active (Figure 3). Such a mechanism would control NURR1 over-activation.

The effect of NURR1 phosphorylation by GSK-3β might be interpreted as the opposite action with other kinases that regulate its stability. Jo et al. reported in 2009 that the Ser/Thr protein kinase AKT phosphorylates NURR1 at Ser347, tagging it for degradation through the UPS (Table 2); meanwhile, one NURR1^AKT^ mutant, resistant to degradation, increased the differentiation of neural precursors to DAergic neurons [15]. However, the relevance of AKT leading to the degradation of NURR1 is not clear, because AKT is a survival kinase that should protect NURR1 (Figure 3). As a matter of fact, AKT phosphorylates GSK-3α and GSK-3β in Ser21 and Ser9, respectively, in their pseudosubstrate domain, leading to its inhibition [103]. Therefore, signals that activate AKT would likely stabilize NURR1 by blocking GSK-3.

Another conflicting aspect of the NURR1 phosphorylation in Ser347 by AKT kinase is that this same serine residue is phosphorylated by RSK and MSK (Figure 3) [93]. The same has been reported for NUR77, where both kinases can phosphorylate this protein at the equivalent serine residue (Ser354). Although the RSK effect on NURR1 has been hardly evaluated, NUR77 phosphorylation was demonstrated in vivo with the use of a specific RSK inhibitor and connected to the effect of intracellular translocation during apoptosis in T lymphocytes. In addition, NUR77 can interact physically with RSK when T lymphocytes are stimulated [93]. On the other hand, phosphorylation of NUR77 of AKT was demonstrated only in vitro [73,97], and there is no clear experimental basis to support the concept that NUR77 can interact with the AKT kinase in T lymphocytes [74]. Additional support for this idea rests on the fact that NUR77 is not altered by treatment with two inhibitors that block AKT activation. Besides, active recombinant AKT versions cannot phosphorylate NUR77 in vitro [73]. Together, these observations suggest that AKT does not have a physiological role in changing the activity of NUR77.

NURR1 participates in the repair of DNA double-strand (dsDNA) breaks. By analogy with NUR77, NURR1 interacts with DNA-PK, which phosphorylates NURR1 at Ser337 in the conserved NR4A sequence “TDSLKG” (Table 2). Overexpression of a Ser337Ala NURR1-variant was a hindering factor in dsDNA repair, demonstrating its involvement in the process [92]. This NURR1 activity was consistent with results in the human severe combined immunodeficiency, where one DNA-PKcs mutant that could not phosphorylate NURR1.

Another well-known function of the NURR1 is related to its capacity to act as a regulator of inflammation [79]. Saijo and collaborators demonstrated that the NLK kinase (Nemo-like kinase) can phosphorylate NURR1 and control its capacity to inhibit NF-κB target genes. The authors did not identify the specific residues that are phosphorylated by NLK, but they speculated that the region would be nearby, “Core2”. The proposed model does not seem to demonstrate that GSK-3β would have an effect on the NURR1 stability by phosphorylation, as mentioned previously. One explanation might be through NLK activity on NURR1 phosphorylation in the same region as GSK-3β, having an opposite effect on the regulation of neuroinflammation. The relationship between GSK-3β and NLK was already established in 2009 [79], as both proteins can regulate the MAP1B activity by phosphorylation in the same amino acid residues. It is likely that a similar mechanism might regulate NURR1, although in this case, NLK might phosphorylate NURR1, then protect against GSK-3β-mediated degradation.

## 5. Regulation of NURR1 by SUMOylation

SUMOylation consists on the reversible covalent attachment of one or multiple SUMO (small ubiquitin-related modifier) proteins to one or several lysine residues of a target protein [104]. SUMO is covalently linked to its substrates through amide (isopeptide) bonds formed between its C-terminal glycine residue and the ɛ-amino group of internal lysine residues. These enzymes are like those involved in the reversible conjugation of ubiquitin. SUMO modification plays important roles in diverse processes such as chromosome segregation and cell division, DNA replication and repair, nuclear protein import, protein targeting to and formation of certain subnuclear structures, and the regulation of a variety of processes, including the inflammatory response in mammals.

Nowadays, NR4A family orphan receptors are known to be regulated by SUMOylation [105,106]. The first evidence for SUMOylation in NURR1 was demonstrated by Galleguillos et al. in the year 2004 [23] (Figure 4). Using in vitro pulldown and immuno-precipitation assays, the authors demonstrated that PIASγ, a SUMO-E3 ubiquitin-protein isopeptide ligase, interacts with NURR1. This interaction caused a reduction in NURR1 transcriptional activity on the monomer response element of the NR4A subfamily (NBRE, 5′-AAAGGTCA-3′). Two putative consensus SUMOylation sites were identified in NURR1, lysine 91 and 577 (Table 2, Figure 4) [23]. Mutations of lysine 91 or 577 to arginine abrogated the ability to be modified by SUMO proteins (Figure 4). Interestingly, PIASγ increased with the addition of SUMO2 protein on NURR1, while a PIASγ mutant lacking SUMO ligase activity was not able to do so, suggesting that PIASγ is important for SUMOylation on these residues [23]. PIASγ and NURR1 are co-localised in the substantia nigra of the mouse midbrain and may have a regulatory effect on the maturation and maintenance of the DAergic phenotype or the onset or progression of neuro-pathologies. Thus, the NURR1 modulation by PIASγ is double: on one side, through its E3 ligase activity and on the other, a putative transcriptional corepressor. The subcellular localization and stability of NURR1 were not affected by SUMOylation at lysine 91 of NURR1 [22].

NURR1 SUMOylation may have physiological consequences [22]. The NURR1 C-terminus (Table 2) can be SUMOylated by SUMO-2 [23] and SUMO-3 [22], as previously demonstrated (Figure 4). Mutations of lysine 558 (belonging to a putative SUMO acceptor site) and 576 in mouse NURR1 (the equivalent of lysine 577 in human NURR1) decreased NURR1 transcriptional activity on the monomer response element NBRE, whereas NURR1 transcriptional activity on the direct response element 5 (DR-5) increased in the presence of the lysine 558 mutation [79]. NURR1 SUMOylation is necessary to recruit the corepressor CoREST to NF-κB target genes, which reduces the expression of proinflammatory genes such as the one encoding Tumor Necrosis Factor-alpha (TNFα), acting like a transrepressor (Figure 4) [79]. This mechanism could be relevant to understanding NURR1 involvement in neuroinflammatory processes [10,79]. Nevertheless, another study demonstrated that the mutation of lysine 558 to arginine did not alter NURR1 SUMOylation [107]. This study suggested that this site might not be a target of SUMOylation, but rather that this mutation could modify the recruitment of co-factors by NURR1 and, therefore, induce changes in NURR1 transcriptional activity. These experiments were done by ectopically expressing NURR1, SUMO-2, and PIASγ, which could lead to non-physiological effects.

## 6. Regulation of the Nuclear-Cytoplasmic Distribution of NURR1

Nucleocytoplasmic transport is another mechanism that regulates NURR1 function. [18]. Efficient protein transport through the nuclear pore complex is crucial for the exchange of proteins between the nucleus and cytoplasm [108]. Multiple proteins, including various transcription factors, histones, and cell cycle regulators, require nuclear localization signals (NLS) or nuclear export signals (NES) [109]. The importing and exporting family of proteins recognizes the sequences found in macromolecules through their soluble transport receptors [110]. The transport signals that interact with importin-α (Imp-α), importin-β (Imp-β), CRM1 (chromosome region maintenance 1, also known as exportin1 or Xpo1), and transportin-1 (also known as karyopherin-β2) are well described [111,112,113,114,115].

Ojeda et al. described the subcellular distribution of NURR1 in adult rat brains. Hybridization was detected in cells across the brain with intense signals in the neurons of the striatum and substance nigra [116,117]. Subsequently, the first evidence of these signal sequences of change of subcellular localization in NURR1 was reported by García-Yagüe et al. in 2013 [18]. The authors found that NURR1 contains a bipartite NLS within its DNA-binding domain (DBD) and two leucine-rich NES in its ligand-binding domain (LBD), which regulate NURR1 shuttling in and out of the nucleus (Table 1). Both NLSs in wild-type NURR1 (NLS1 and NLS2) control the nuclear localization of NURR1 (Figure 5). The diverse biological activities of NURR1 depend on their subcellular localization. In the nucleus, NURR1-mediated transcription is generally implicated in differentiation and survival [18,118], but NURR1 can be also exported out of the nucleus. Translocation of NURR1 from the nucleus to the cytoplasm has been reported in post-ischemic striatal cells preceding pyknosis [119], and in vitro PD models based on redox dysregulation using 6-OHDA [120], calcium influx, or glutamate excitotoxicity [121]. According to García-Yagüe et al. (2013), NURR1 presents two NES (NES1 and NES2) that control its translocation from the nucleus to the cytoplasm, and are localized to the end-carboxyl terminal into the ligand-binding domain (LBD) [18]. One of them (NES2) is redox-sensitive [18], because treatment with the oxidative reagent sodium arsenate induced its transport from the nucleus to the cytoplasm regulated by the CRM1 exporting (Figure 5). A NES2 mutant was more insensitive to export than the wild-type NURR1 [18]. Therefore, it was suggested that NURR1 can regulate its nucleus-cytoplasmic transition according to the demands in redox balance, involving a new mechanism to control NURR1 transcriptional activity and the subsequent DAergic phenotype under stress conditions [18,122].

## 7. Concluding Remarks

NURR1 plays a fundamental role in the differentiation and maintenance of the DAergic phenotype, cytoprotective functions, and other physiological processes. Post-translational modifications of NURR1 catalysed by several protein kinases and SUMO E3-ligases, as well as protein-protein associations are critical for the regulation of its stabilization and transcriptional activity. Regulation of NURR1 by phosphorylation is one of the most controversial aspects. Thus, NURR1 is degraded through the ubiquitin-proteasome system (UPS) by GSK-3β and AKT. The “Core 2” motif of NURR1 is apparently targeted by GSK-3β and ERK1/2. This motif is also targeted by RSK, downstream of AKT. Therefore, the mechanisms underlying how specific kinases context-dependently regulate the NURR1, the entire specific phosphorylation sites of some kinases, and the whole landscape of NURR1 phosphorylation regulation still require further exploration.

The restoration or enhancement of NURR1 activity may prevent the onset and progression of several pathologies. Even though research on the mechanisms of NURR1 regulation began more than 20 years ago, it is still not possible to translate the progress on the physio-pathological mechanism of NURR1 from animal models to clinical trials. NURR1 regulation by heterodimerization with RXR is one exception that appeared in a preclinical translational study in mouse PD models [123], using pharmacological approaches that regulate and improve its activity, either onset or impairment progression of the physio-pathological condition [18,69,70,71]. Therefore, it is necessary to evaluate which are the physiological implications on the different diseases, including neural disorders, cancer, metabolic diseases, inflammation impairment, and others. Currently, the exact mechanisms underlying the control of NURR1 activity and its impact on psychopathology still require further exploration.

## Figures and Tables

**Figure 1 ijms-24-12280-f001:**
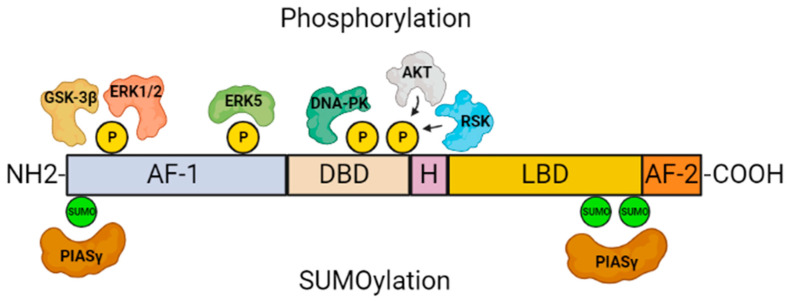
Domain structure of NURR1 and post-translational modification sites. Schematic representation of NURR1 indicating phosphorylation and SUMOylation domains. AF: activation function; DBD: DNA-binding domain; H: hinge; LBD: ligand-binding domain.

**Figure 2 ijms-24-12280-f002:**
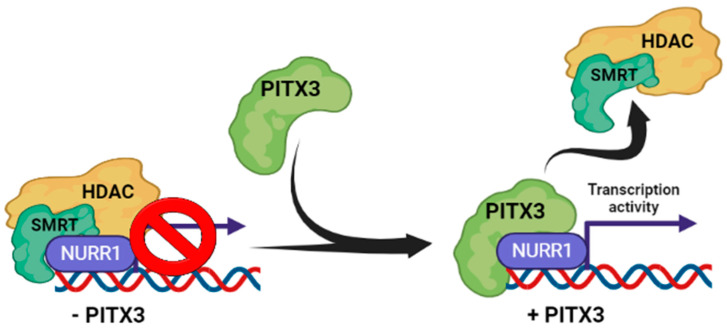
PITX3 regulates NURR1–transcription activity. In the absence of PITX3, NURR1 is in a state of transcriptional repression by binding to SMRT, which recruits a histone deacetylase (HDAC), and keeps the promoter repressed. The recruitment of PITX3 to NURR1 promoters allows the uncoupling of the SMRT/HDAC repressor complex, which favors the complete transcriptional activation of NURR1.

**Figure 3 ijms-24-12280-f003:**
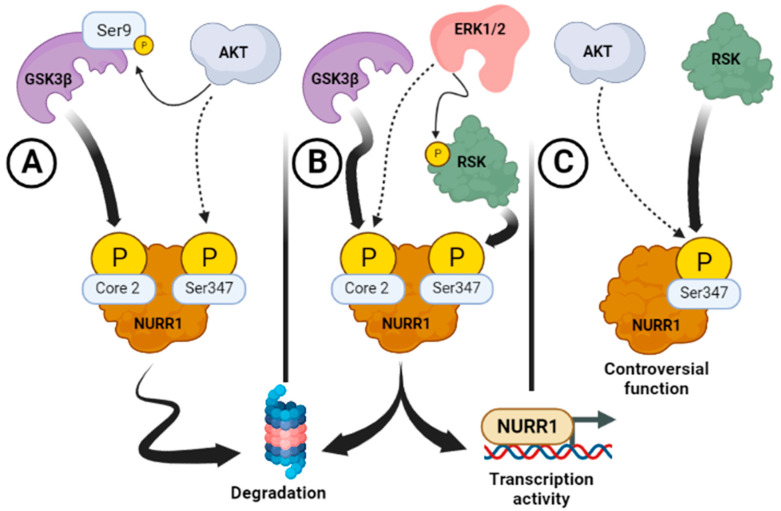
Regulation of NURR1 by phosphorylation. Schematic representation of the different situations for NURR1 phosphorylation. (**A**) GSK-3β and AKT compromise NURR1 stability and lead to its degradation by the ubiquitin-proteasome system. However, the relevance of AKT leading to the degradation of NURR1 is not clear considering that AKT is a survival kinase that should protect NURR1 (dashed arrow). AKT phosphorylates GSK-3β at Ser9 in its pseudosubstrate domain, leading to its inhibition (lineal thin arrow). Therefore, signals that activate AKT will likely stabilize NURR1 through the inhibition of GSK-3β. (**B**) ERK1/2 and GSK-3β are two kinases that are active/inactive in the presence/absence of growth factors, respectively. In these cases, the NURR1 “Core2” might function as a molecular switch in coordination with other signals, leading to NURR1 activation or proteasomal degradation. Many GSK-3β substrates need to be previously phosphorylated by another kinase in order to be recognized by GSK-3β. Therefore, it is possible that the initial phosphorylation, originated by ERK (dashed arrow), might lead to transient NURR1 activation and, at the same time, that ERK-phosphorylated NURR1 would be primed for degradation at a later stage when cell signaling is reduced and GSK-3β becomes active. Another possibility is that ERK1/2 might regulate NURR1 activity through other downstream kinases (lineal thin arrow). In this case, RSK would be the intermediate kinase that might regulate NURR1 activity (lineal thick arrow). (**C**) NURR1 phosphorylation has been reported to occur by AKT at Ser347 leading to decreased protein stability (dashed arrow); meanwhile, RSK/MSK can phosphorylate at the same residue (lineal thick arrow). Although the RSK/MSK effect is unknown in NURR1, these pathways promote cell survival, further contradicting the activity of AKT as a propeller of NURR1 degradation. Dashed arrow (doubtful effect), Lineal thin arrow (consensual effect), Lineal thick arrow (proven effect).

**Figure 4 ijms-24-12280-f004:**
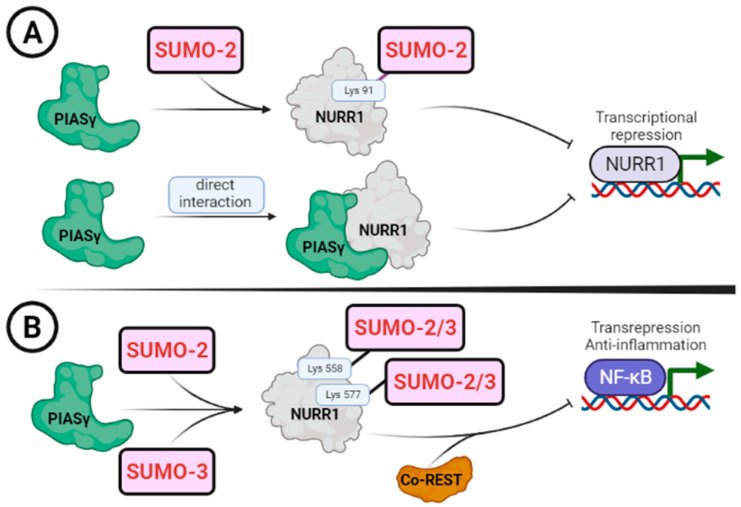
Effect of SUMOylation on NURR1 activity. Schematic representation of the different conditions for NURR1 SUMOylation. (**A**) NURR1 is SUMOylated by PIASγ, adding one molecule of SUMO-2 in the Lys-91, located in the transcriptional activation function 1 domain of NURR1. Lys-91 is the major target of NURR1 SUMOylation because it contains the canonical SUMOylation motif. Moreover, PIASγ fully represses NURR1 transactivation through direct interaction, independently of its E3-ligase activity. (**B**) NURR1 has also been SUMOylated in two other lysine residues (K558 and K577) by PIASγ. Both K558 and K576 are located in the ligand-binding domain and close to the I-box, and are required for repression of NF-κB activity. SUMOylation in K558 and K576 leads to recruitment of the CoREST repressor. NURR1/CoREST-mediated transrepression complex blocks NF-κB p65 activity.

**Figure 5 ijms-24-12280-f005:**
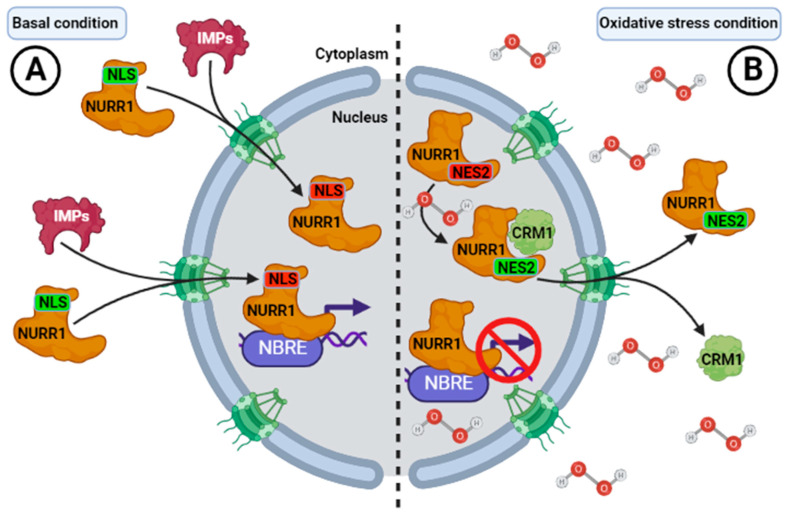
Control of the NURR1 by subcellular localization. Schematic representation of the elements that control the trafficking of NURR1 between cytoplasm and nucleus. (**A**) NURR1 is mainly a nuclear protein and presents nuclear localization signals (NLS) that are recognized specifically by cargo cytoplasmic proteins named importins (IMPs). NURR1 possess one nuclear localization signal bipartite (termed here NLS), NLS1 conforms to the consensus structure of typical NLS (ZZX(10–20)ZZZ, where Z is Lys or Arg and X is any amino acid), and other atypical NLS2 consisting of scattered Arg and Lys residues in a small stretch of 12 amino acids at positions 338–350, also in the DNA-binding domain. (**B**) NURR1 have two nuclear export signals (NES) participating in its translocation from the nucleus to the cytoplasm, and are recognized by cargo proteins, including exportin-1 (CRM1). Both sequences conform to the consensus motif LX(1–3)LX(2–3)LXL, where L is Leu and X is any amino acid. One of them, NES2, is specifically important because it participates in the redox-dependent export of NURR1.

**Table 1 ijms-24-12280-t001:** Partner binding regulators of NURR1.

Partners	Partner-Binding Domain	NURR1-Binding Motif	Effect	Reference
RXRα; RXRγ	I-box (LBD) Glu^390^, Glu^394^	I-box (LBD) Lys^554^-Leu^555^-Leu^556^	Transcription activity	[21,60]
GR	Full-length protein (A/B, DBD and LBD)	Transactivation domain A/B (first 58 aas)	Transcription activity	[19]
PITX3	Unknown	Unknown	Transcription activity	[73]
p57Kip2	Unknown	Transactivation domain A/B (Full-length)	Transcription activity	[74]
SRCs	PAS-B domain	AF1 domain (Transactivation A/B)	Transcription activity	[75]
PIN1	WW domain	Transactivation domain A/B and DBD	Transcription activity	[76]
FHL2	Full-length protein (LIMs domains)	Transactivation domain A/B and DBD	Transcription activity	[77]
RUNX1	Unknown	Unknown	Differentiation CD4^+^ T cells	[78]
CoREST	Unknown	DBD	Transrepression	[79]
p53	COOH-terminal	DBD	Anti-apoptotic	[80]
Importins	Cargo-recognition motif	DBD (NLS)	Import to nucleus	[18]
Exportins (CRM1)	Cargo-recognition motif	LBD (NES)	Export to nucleus	[18]

**Table 2 ijms-24-12280-t002:** Post-translational modifications reported for NURR1.

**Kinase**	**Phospho-Motifs**	**Effect**	**Reference**
ERK1/2	124-PS***pS(126)***PPTPS***pT(132)***P-133	Transcription activity	[91]
GSK-3β	124-PS ***pS(126)***PP***pT(129)***PS***pT(132)***P-133	Degradation	[16]
ERK5	166-RK***pT(168)***PVSRLSLF***pS(177)***FK-179	Transcription activity	[83]
DNA-PK	335-TD***pS(337)***LKG-340	Double-strand break repair	[92]
AKT	344-LP***pS(347)***KP-349	Degradation	[15]
RSK	344-LP***pS(347)***KP-349	Unknown	[93]
**SUMO-E3 Ligases**	**SUMO-Motifs**	**Effect**	**Reference**
PIASγ/PIAS4	85-GQQSSI**K^SUMO-2^(91)**VEDIQMH-98	Transcription repression	[22,23]
PIASγ/PIAS4	571-QRIFYL**K^SUMO-2/3^(577)**LEDLVPP-584	Transcription repression	[23]
PIASγ/PIAS4	552-LSKLLG**K^SUMO-2/3^(558)**LPELRTL-565	Transrepression	[23,79]

## Data Availability

Data are contained within the article.

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
