# Peer review of "Mechanisms of NURR1 Regulation: Consequences for Its Biological Activity and Involvement in Pathology"

_ijms, 2023, doi:10.3390/ijms241512280_

Round 1

Reviewer 1 Report

This review summarizes the regulation mechanism of nuclear receptor-related protein (NURR1) and related biological consequences. The authors mainly focused on some important pathological significances and were able to lay out an excellent story by providing up-to-date progress in the field. Even though the scientific portion of the article is very good, it requires thorough editing before publishing. A couple of sentences in the manuscript are very difficult to understand, and some terms like “Core 2” are inconsistent.

 Some citations are missing, for example, “Although the effect of ERK1/2 on NURR1 phosphorylation seems compelling, these in vitro results do not seem to impact the biological activity in cell assays.”

Replacing mono and bidimensional gels with 1D and 2D will make it easy to follow.

The authors didn’t mention much about how alpha-synuclein and NURR1 regulation.

A cartoon representation of SUMOylation and oligomerization will help to understand the mechanism. 

requires editing

Author Response

Dear Review #1

All their comments have been addressed as follows:

This review summarizes the regulation mechanism of nuclear receptor-related protein (NURR1) and related biological consequences. The authors mainly focused on some important pathological significances and were able to lay out an excellent story by providing up-to-date progress in the field. Even though the scientific portion of the article is very good, it requires thorough editing before publishing.

ANSWER: Thank you for these positive comments. The manuscript has been revised to improve English language and scientific expressions.

A couple of sentences in the manuscript are very difficult to understand, and some terms like “Core 2” are inconsistent.

ANSWER: we hope that the extensive editing that we did on the entire manuscript will have solve this issue. Regarding “Core 2” we have given a more detailed explanation and it is now also explained in Figure 2.

Some citations are missing, for example, “Although the effect of ERK1/2 on NURR1 phosphorylation seems compelling, these in vitro results do not seem to impact the biological activity in cell assays.”

ANSWER: This and other new references have been added.

Replacing mono and bidimensional gels with 1D and 2D will make it easy to follow.

ANSWER: Changed. Thank you.

The authors didn’t mention much about how alpha-synuclein and NURR1 regulation.

ANSWER: We have included a brief description of the relationship between alpha-synuclein and NURR1 in the GSK3 activity context. Bottom of page 6

A cartoon representation of SUMOylation and oligomerization will help to understand the mechanism. 

ANSWER: Thanks for this suggestion. The new Figure 4 for SUMOylation and Figure 2 for Oligomerization provides this cartoon.

Reviewer 2 Report

García-Yagüe and Antonio Cuadrado present here a very detailed review on the regulation of the transcription factor NURR1. The review is thorough and the basic content good, however there are a number of structural issues with how it is written that need to be addressed before it could be published.

The biggest of these is that, while I understand the authors want to focus on regulation of NURR1, rather than write a general review of NURR1, there simply is not enough introduction to the protein to establish what it is, or why its regulation is important. Please can the authors greatly expand chapter 1 of the review (likely into two chapters, one overviewing its function, the other its implications for disease.  Again, while I understand that this is not the focus of the review, it is absolutely necessary as a foundation for readers less familiar with NURR1, and so that the rest of the review makes sense.

The second is that the authors take a very narrative, almost critical approach, when they talk through individual experiments papers in great depth, with comments on the value of different approaches. This makes it very hard to read and understand the overall picture of NURR1 regulation. A more concise approach, attempting to summarize information for ease of understanding, would seem to be much more valuable.

Other points:

A diagram of the domain structure of NURR1, indicating the location of key post-translational modifications, would be extremely helpful.  This would probably be a one-dimensional schematic, although showing any solved 3D structure in addition would also be good.

It seems strange that the phosphorylation section (chapter 2), which features phosphorylation-dependent protein-protein interactions, comes before the chapter on protein-protein interactions (chapter 4)

There are a lot of unusual grammatical structures that make the manuscript hard to read in places. Some examples from the first few pages include:

1. The changing between SUMOrylation (with an r) and SUMOylation (without an r). The latter is preferred

2. Abstract: “affecting to the expression” > “affecting the expression”

3. Abstract: “we compile nowadays all” > “we summarize all currently known”

4. Putting “in vitro” in practices on some occasions, but not others.

5. Referring to authors in the format “Surname Initial”, when just “Surname” or “Initial Surname” would be more common.

6. Various occasions: Using “the bibliography” when they should say “the scientific literature” or just “the literature”

7. Bottom of page 7: “exists incoherent” should maybe be “reveals contradictory information”

8. Start of page 3: “Besides” > “Additionally”

9. Half way through page 3: “explication” to “explanation”

9. Late on page 3: “the authors showed that not just NUR77 is phosphorylating by them, such as NURR1” > “the authors show that other proteins besides NUR77, such as NURR1, are also phosphorylated by this kinase”

10. Bottom of page 3: “being able to suggest control of the balance subcellular localization into the cell” > “suggesting active regulation of its subcellular localization”

etc

Author Response

Dear Review #2

All their comments have been addressed as follows:

García-Yagüe and Antonio Cuadrado present here a very detailed review on the regulation of the transcription factor NURR1. The review is thorough and the basic content good, however there are a number of structural issues with how it is written that need to be addressed before it could be published.

The biggest of these is that, while I understand the authors want to focus on regulation of NURR1, rather than write a general review of NURR1, there simply is not enough introduction to the protein to establish what it is, or why its regulation is important. Please can the authors greatly expand chapter 1 of the review (likely into two chapters, one overviewing its function, the other its implications for disease.  Again, while I understand that this is not the focus of the review, it is absolutely necessary as a foundation for readers less familiar with NURR1, and so that the rest of the review makes sense.

ANSWER: Thank you for this comment. We have split the introduction in two sections. One is the introduction itself explaining the structural and functional role of NURR1. The other is related to the participation of NURR1 in several pathologies.

The second is that the authors take a very narrative, almost critical approach, when they talk through individual experiments papers in great depth, with comments on the value of different approaches. This makes it very hard to read and understand the overall picture of NURR1 regulation. A more concise approach, attempting to summarize information for ease of understanding, would seem to be much more valuable.

ANSWER: in the new version we have tried to reduce details and give more focus on the overall picture.

Other points:

A diagram of the domain structure of NURR1, indicating the location of key post-translational modifications, would be extremely helpful.  This would probably be a one-dimensional schematic, although showing any solved 3D structure in addition would also be good.

ANSWER: Thanks for this helpful comment. The new Figure 1 provides a one-dimensional esquematic.

It seems strange that the phosphorylation section (chapter 2), which features phosphorylation-dependent protein-protein interactions, comes before the chapter on protein-protein interactions (chapter 4)

ANSWER: we agree with the reviewer that protein-protein interactions might go first. We have changed the order.

Comments on the Quality of English Language

There are a lot of unusual grammatical structures that make the manuscript hard to read in places.

ANSWER: English style has been revised along the entire manuscript.

Some examples from the first few pages include:

  1. The changing between SUMOrylation (with an r) and SUMOylation (without an r). The latter is preferred

ANSWER: Changed.

  1. Abstract: “affecting to the expression” > “affecting the expression”

ANSWER: Changed.

  1. Abstract: “we compile nowadays all” > “we summarize all currently known”

ANSWER: Changed.

  1. Putting “in vitro” in practices on some occasions, but not others.

ANSWER: Changed to in vitro or in vivo.

  1. Referring to authors in the format “Surname Initial”, when just “Surname” or “Initial Surname” would be more common.

ANSWER: Changed to Surname.

  1. Various occasions: Using “the bibliography” when they should say “the scientific literature” or just “the literature”

ANSWER: Changed.

  1. Bottom of page 7: “exists incoherent” should maybe be “reveals contradictory information”

ANSWER: Changed to “contradictory information”.

  1. Start of page 3: “Besides” > “Additionally”

ANSWER: Changed.

  1. Half way through page 3: “explication” to “explanation”

ANSWER: Changed.

  1. Late on page 3: “the authors showed that not just NUR77 is phosphorylating by them, such as NURR1” > “the authors show that other proteins besides NUR77, such as NURR1, are also phosphorylated by this kinase”

ANSWER: Changed.

  1. Bottom of page 3: “being able to suggest control of the balance subcellular localization into the cell” > “suggesting active regulation of its subcellular localization”

etc

ANSWER: Changed.

Round 2

Reviewer 2 Report

The authors have addressed all of my significant concerns, and I recommend the article for publication.  Please note however that, while the authors have corrected the specific English mistakes I pointed out, further work is needed in this regard.

The authors addressed all of my specific English corrections that I gave them previously, however these were all taken from the first 3 pages, and supposed to be examples, not evidence of everything that needed doing. I give a few more examples from chapter 6 here, but again these are only examples:

“The importins and exportins family of…” to “The importin and exportin family of…”

“Hybridization was detected in scattered cells…” to “Hybridization was detected in a wide array of cells…” or ““Hybridization was detected in cells across the brain…” (depending on author’s meaning)

Definitions of NLS and NES given multiple times.

“Both NLSs in NURR1 naïve…” is very unclear, maybe they mean “Both NLSs in wild-type NURR1 …”

“…because the treatment with the oxidative reagent sodium arsenate…” to “because treatment with the oxidative reagent sodium arsenate” (no “the”)

Author Response

Dear Reviewer #2,

We thank the reviewer #2 for his/her kind comments and for providing input to improve the quality of this review. All his/her comments have been addressed as follows:

Comments and Suggestions for Authors

The authors have addressed all of my significant concerns, and I recommend the article for publication.  

ANSWER: Thank you, we hope that the review will be useful to those working or starting to work on NURR1.

Please note however that, while the authors have corrected the specific English mistakes I pointed out, further work is needed in this regard.

ANSWER: English style has been revised again along with the entire manuscript.

Comments on the Quality of English Language

The authors addressed all of the specific English corrections that I gave them previously, however, these were all taken from the first 3 pages, and supposed to be examples, not evidence of everything that needed doing.

ANSWER: The English has been meticulously corrected paragraph by paragraph on all pages with the assistance of grammar and vocabulary correction software. In the review, we noticed specific errors, which we have also corrected.

I give a few more examples from chapter 6 here, but again these are only examples:

 “The importins and exportins family of…” to “The importin and exportin family of…”

ANSWER: Changed.

“Hybridization was detected in scattered cells…” to “Hybridization was detected in a wide array of cells…” or ““Hybridization was detected in cells across the brain…” (depending on author’s meaning)

ANSWER: Changed. We prefer to choose, ““Hybridization was detected in cells across the brain…”

Definitions of NLS and NES given multiple times.

ANSWER: We have left the definition of NLS and NES only the first time that they are mentioned.

“Both NLSs in NURR1 naïve…” is very unclear, maybe they mean “Both NLSs in wild-type NURR1 …”

ANSWER: Changed.

“…because the treatment with the oxidative reagent sodium arsenate…” to “because treatment with the oxidative reagent sodium arsenate” (no “the”)

ANSWER: Changed.

We thank again, these useful comments and hope that this time you will find the manuscript suitable for publication.

Best regards,

Angel Juan García-Yagüe
